# Evaluation of an Antibiotic Susceptibility Testing Method on *Enterobacterales*-Positive Blood Cultures in Less Than 8 h Using the Rapid Mueller-Hinton Diffusion Method in Conjunction with the SIRscan 2000 Automatic Reading Device

**DOI:** 10.3390/microorganisms10071377

**Published:** 2022-07-08

**Authors:** Mathilde Payen, Alice Gaudart, Kevin Legueult, James Kasprzak, Audrey Emery, Grégoire Mutambayi, Christian Pradier, Frédéric Robin, Romain Lotte, Raymond Ruimy

**Affiliations:** 1Laboratoire de Bactériologie, Hôpital L’Archet 2, Centre Hospitalier Universitaire de Nice, 06000 Nice, France; payen.m@chu-nice.fr (M.P.); gaudart.a@chu-nice.fr (A.G.); james@pacalab.fr (J.K.); emery.a@chu-nice.fr (A.E.); mutambayi.g@chu-nice.fr (G.M.); lotte.r@chu-nice.fr (R.L.); 2Institut National de la Santé et de la Recherche Médicale U1065, Centre Méditerranéen de Médecine Moléculaire, Équipe 6, 06200 Nice, France; 3Faculté de Médecine, Université Côte d’Azur, 06200 Nice, France; 4Département de Santé Publique, UR2CA, Université Côte d’Azur, Centre Hospitalier Universitaire de Nice, 06000 Nice, France; legueult.k@chu-nce.fr (K.L.); pradier.c@chu-nice.fr (C.P.); 5Laboratoire de Bactériologie Clinique, Institut National de la Santé et de la Recherche Médicale U1071, INRA USC2018, Centre Hospitalier Universitaire de Clermont-Ferrand, Université Clermont Auvergne, 63000 Clermont-Ferrand, France; frobin@chu-clermontferrand.fr; 6Centre National de Référence de la Résistance aux Antibiotiques, Laboratoire Associé, 63000 Clermont-Ferrand, France

**Keywords:** antibiotic susceptibility testing, rapid AST, bacteremia

## Abstract

*Enterobacterales* bloodstream infections are life-threatening and require rapid, targeted antibiotherapy based on antibiotic susceptibility testing (AST). A new method using Muller-Hinton Rapid-SIR (MHR-SIR) agar (i2a, Montpellier, France) allows complete direct AST (dAST) to be read from positive blood culture bottles (BCBs) for all *Enterobacterales* species after 6–8 h of incubation. We evaluated (i) the performance of dAST from positive BCBs on MHR-SIR agar using two different inoculum protocols; (ii) the categorical agreement between dAST results obtained with MHR-SIR agar vs. those obtained with Muller-Hinton (MH) agar; and (iii) the ability of the MHR-SIR medium to detect β-lactam resistant *Enterobacterales.* Finally, we estimated the saved turnaround time (TAT) with MHR-SIR compared with MH agar in our 24/7 laboratory. Our results showed that the most suitable inoculation protocol for dAST on MHR-SIR agar was 1 drop of BCB/5 mL H_2_O. For monomicrobial *Enterobacterales* BCBs, dAST performed on MHR-SIR medium showed 99.3% categorical agreement with AST on MH agar. Furthermore, MHR-SIR agar allows early detection of β-lactam resistance mechanisms, including AmpC hyperproduction, extended-spectrum β-lactamase, and carbapenemase. Finally, TAT reduction in our 24/7 laboratory was 16 h, enabling a significantly faster provision of antibiotic advice.

## 1. Introduction

Sepsis is a major public health issue due to high mortality rates, which are estimated at 10% among children and 38.4% among the elderly [1]. *Enterobacterales* are the organisms most frequently responsible for sepsis [2]. The greater mortality in bloodstream infections is due to (i) delayed antibiotic administration [3], (ii) inappropriate antibiotic therapy [4,5,6], or (iii) the presence of antibiotic resistance, such as extended-spectrum β-lactamase (ESBL) [7]. Antibiotic susceptibility testing (AST) is performed in order to determine which specific antibiotic a particular bacterium isolated from blood culture bottles (BCBs) is sensitive to, thus allowing therapeutic adjustment (antibiotic escalation or de-escalation). Rapid AST results are associated with improved clinical outcomes in sepsis patients [8]. Currently, AST performed on Muller-Hinton (MH) agar from positive BCBs can be interpreted for all *Enterobacterales* and antibiotics only after 16 to 24 h of incubation. In the last few years, microbiologists have sought to reduce the AST turnaround time (TAT) for these critical patients. In 2020, the European Committee on Antimicrobial Susceptibility Testing (EUCAST) published guidelines for *Escherichia coli* and *Klebsiella pneumoniae* testing in 4 to 8 h [9]. In the same year, the Clinical and Laboratory Standards Institute (CLSI) released guidelines for all *Enterobacterales* AST testing in 8 to 10 h, but only for six antibiotics [10]. These two guidelines are of interest, but they do not encompass several *Enterobacterales* species with multiple antibiotics. 

A disc diffusion AST method for all *Enterobacterales* species has been recently developed, which allows a large panel of antibiotics to be read from 6 to 8 h after incubation. Mueller-Hinton Rapid-SIR (MHR-SIR) agar (i2a, Montpellier, France) would allow a reduction in AST TAT of up to 16 h compared with ASTs read after 16 to 24 h of incubation [11]. Studies have been carried out to evaluate the MHR-SIR medium following inoculum recommendations issued by the British Society for Antimicrobial Chemotherapy (BSAC) [11,12]. In 2018, the Comité de l’Antibiogramme de la Société Française de Microbiologie (CASFM), which is associated to the European Committee on Antimicrobial Susceptibility Testing (CASFM-EUCAST), published inoculum recommendations for performing direct AST from positive BCBs [9]. 

In the present study, we first compared the performance of AST using MHR-SIR agar from positive BCBs spiked with quality control *E. coli* strain ATCC 25922 using two different inoculum concentrations, one recommended by the BSAC [12], the other by CASFM-EUCAST [9], to determine which gives the most reproducible and repeatable results. We then prospectively evaluated the categorical agreement between AST using MHR-SIR agar and AST using standard MH agar for a large panel of *Enterobacterales*-positive BCBs obtained from hospitalised patients with sepsis. We also assessed the ability of the MHR-SIR medium to detect multi-drug resistance of *Enterobacterales*. Finally, we estimated the reduction in TAT in our 24/7 laboratory, where AST interpretation was performed from 8:30 a.m. to 6:00 p.m.

## 2. Material and Methods 

### 2.1. Laboratory Setting and Blood Culture Procedure

This study was conducted in our 24/7 bacteriology laboratory at the Nice Teaching Hospital (1800-bed tertiary care centre). The laboratory processes around 65,000 BCBs annually, which represents 49% of the analyses it performs. All BCBs are incubated in a BacT/ALERT^®^ 3D automated device (BioMérieux^®^, Marcy l’Etoile, France) for up to 5 days until positive results are detected. Positive blood cultures undergo Gram staining (PREVI^®^ Color automated Gram staining system; BioMérieux^®^) and direct identification by MALDI-TOF MS (MicroFlex LT and Biotyper database; Brüker, Wissembourg, France) as previously described [13]. In our standard protocol, AST is performed by direct inoculation by swab [13] from the positive BCB onto the MH agar according to CASFM-EUCAST inoculum recommendations (15 drops of culture broth in 9 mL of 0.9% NaCl), followed by incubation for 16 to 18 h in a SIRscan 2000 Automatic incubator-reader (i2a, Montpellier, France). The SIRscan 2000 Automatic includes an expert system that categorises each antibiotic according to CASFM-EUCAST breakpoints (2019 v2) [14].

In our study, susceptibility to the following 18 antibiotics was tested: amoxicillin 20 µg (AM20), amoxicillin-clavulanic acid 30 µg (AMC30), ticarcillin 75 µg (TIC75), ticarcillin-clavulanic acid 85 µg (TIM85), piperacillin 30 µg (PRL30), piperacillin-tazobactam 36 µg (TZP36), cefotaxime 5 µg (CTX5), ceftazidime 10 µg (CAZ10), cefepime 30 µg (FEP30), aztreonam 30 µg (ATM30), ertapenem 10 µg (ETP10), imipenem 10 µg (IPM10), gentamicin 10 µg (GMN10), amikacin 30 µg (AMK30), tobramycin 10 µg (TOB10), nalidixic acid 30 µg (NA30), ciprofloxacin 5 µg (CIP5), and trimethoprim-sulfamethoxazole 25 µg (SXT25). The ASTs on MH agar were read and interpreted between 8:30 a.m. and 5:00 p.m., at least 16 h after agar inoculation, as recommended by CASFM-EUCAST 2019 v2. 

The ASTs on MHR-SIR agar must be read after 6 h incubation in the SIRscan 2000 Automatic; the device takes a picture every 30 min until the eighth hour of incubation and measures the inhibition disc diameters. The MHR-SIR medium contains adjuvants that allow rapid detection of thin bacterial growth. The adjuvants do not change the appearance of the agar, which looks exactly like a standard MH agar. ASTs on MHR-SIR agar can be performed either from a BCB or from a colony. The study design is described in Figure 1.

### 2.2. Comparison of AST Performance Directly from BCBs on MHR-SIR Agar Using Two Inoculum Protocols Proposed by the BSAC and CASFM-EUCAST, Respectively

Bottles containing 10 mL of sterile human blood were spiked with 1 mL of 0.5 McFarland *Escherichia coli* ATCC 25922 and then incubated in a BacT/ALERT^®^ 3D system. As soon as the BCB flagged positive, direct identification was carried out by proteomics with mass spectrometry (MALDI-TOF) as described [13], and AST was performed on MHR-SIR agar with a standard inoculum of 15 drops of positive blood culture broth diluted in 9 mL of saline solution (NaCl), according to CASFM-EUCAST recommendations [14]. Concurrently, rapid AST was performed on MHR-SIR agar with an inoculation protocol of 1 drop in 5 mL of sterile water, according to BSAC recommendations [12]. A 20 gauge needle was used with both inoculation protocols. 

The results of the ASTs with each inoculation protocol (BSAC and CASFM-EUCAST) were evaluated using the acceptable quality control limits for *E. coli* strain ATCC 25922 described in CASFM-EUCAST 2019 v2 [14] and then compared. Repeatability and reproducibility tests were performed on both sets of results. Repeatability was achieved with 20 ASTs performed on the same day by the same operator, using the same positive BCB previously spiked and incubated as described above, and the same microbiologist. Reproducibility, performed by different operators, was assessed by conducting 33 ASTs over 3 days using the same positive BCB. The diameters were read and corrected, if necessary, by the same microbiologist. Disc diameters and interpretations were extracted from the SIR-scan 2000 Automatic using the SIRWEB software.

For each antibiotic tested during the repeatability and reproducibility assays, the disc diameters obtained following the BSAC or CASFM-EUCAST inoculation guidelines were compared with the acceptable limits for *E. coli* strain ATCC 25922 described in CASFM-EUCAST 2019 v2. The average disc diameters of each antibiotic and the number of values outside the acceptable range obtained for the two inoculation protocols (BSAC and CASFM-EUCAST) were then compared. 

### 2.3. Prospective Comparison of MHR-SIR vs. MH on Enterobacterales-Positive Blood Cultures

The performances of the MHR-SIR and the MH methods were compared prospectively through testing of positive BCBs from patients hospitalised at the Nice Teaching Hospital over a 4-month period (June–September) in 2019 and a 2-month period (July–August) in 2020. All the BCBs with monomicrobial *Enterobacterales* culture that flagged positive between 8:30 a.m. and 6:00 p.m. were included. The two AST methods were performed consecutively directly from the positive BCBs. The MHR-SIR medium was inoculated according to BSAC recommendations (1 drop in 5 mL sterile water), and the MH agar was inoculated according to CASFM-EUCAST recommendations (15 drops in 9 mL NaCl). Disc diameters were interpreted in accordance with CASFM-EUCAST 2019 v2 [14]. The results of the MHR-SIR AST were not communicated to doctors. 

For each bacterium and each antibiotic, disc diameters were determined and classified as either susceptible = S, intermediate = I, or resistant = R. Misclassifications were determined according to the following criteria: (i) minor error (me) if one result was classed as intermediate and the other as either susceptible or resistant, (ii) Major Error (ME) (i.e., false resistant) if one result was classed as R with MHR-SIR and S with MH, (iii) Very Major Error (VME) (i.e., false susceptible) if one result was classed as S with MHR-SIR and R with MH agar. Where there were inconsistencies between the two methods, categorisation was determined by minimum inhibitory concentrations (MICs) and E-test on colonies with a controlled inoculum of 0.5 McFarland (BioMérieux^®^, La Balme-Les-Grottes, France). 

Where AST revealed synergy between clavulanic acid and third-generation cephalosporins (3GC, i.e., cefotaxime, ceftazidime, aztreonam) or cefepime, extended-spectrum ß-lactamase (ESBL) production was identified by the microbiologist. If synergy was not clearly visualised, but an ESBL producer was suspected (i.e., ticarcillin resistant and at least one 3GC categorised as R or cefepime R), a double-disc synergy test (DDST) was performed using amoxicillin-clavulanic acid combined with 4 other antibiotics, cefotaxime, ceftazidime, aztreonam, and cefepime, as previously described [15]. 

### 2.4. Evaluation of MHR-SIR Ability to Detect Multidrug-Resistant (MDR) Enterobacterales

In order to verify the ability of the MHR-SIR medium to detect resistance mechanisms such as ESBL or carbapenemase, we selected and tested 39 strains with various β-lactam resistance profiles. A sterile BCB containing 10 mL of human blood was spiked with 1 mL of 0.5 McFarland *Enterobacterales* strain. The collection comprised 19 ESBL producers, 10 carbapenemase producers, 9 AmpC hyperproducers, and 8 plasmid-mediated cephalosporinase producers, while 5 strains produced other enzymatic resistance mechanisms (Table 1). MICs for carbapenems (imipenem and meropenem), cefoxtaxime, and ceftazidime were determined by E-test on colonies (Appendix A). DNA of each isolate was extracted with a DNeasy UltraClean Microbial Kit (Qiagen, Hilden, Germany). The libraries were prepared with a Nextera XT Kit (Illumina, San Diego, CA, USA), and they were sequenced by the Illumina MiSeq system. The whole-genome sequence (WGS) of each strain was determined by de novo assembly of 2 × 301-bp paired-end reads generated by Illumina technology using assembler SPAdes [16] (average sequencing depth: ≥75×; genome size: 4,567,913–5,210,322 nucleotides) as previously described [17]. The antibiotic resistance genes were characterized as previously reported [17]. 

As described above, ASTs were performed with MHR-SIR medium and the BSAC inoculum protocol, or with standard MH agar and the CA-SFM inoculum protocol. They were incubated in the SIRscan 2000 Automatic for the appropriate length of time and then read by the device. Disc diameters and interpretations obtained with both media were extracted from the SIR-scan 2000 Automatic system using the SIRWEB software and were compared. Where there were discrepancies, categorisation was retrospectively determined by a third method (E-test) performed on colonies. 

The detection strategy for ESBL producers was as described above in the prospective comparison section. AmpC hyper producers and plasmid-mediated cephalosporinase producers were identified when at least one 3GC was categorised as I or R and no synergy was detected by the DDST. Where 3GC were categorised as S and Piperacillin-tazobactam as R, oxacillinase and other penicillinases were deduced and interpreted by a microbiologist. When carbapenemase production was suspected, i.e., when the diameter of the ertapenem disc was categorised as I or R, a Carba-NP carbapenemase screening test was performed [18]. The results of the complementary tests were compared with the reference method to characterise multidrug resistance based on the presence of resistance genes found by WGS of the isolate. 

### 2.5. Estimated Reduction in AST TAT When Using Direct MHR-SIR from POSITIVE Blood Cultures

In this single-centre study, AST TAT was retrospectively estimated between 1 July 2017 and 31 October 2017 on all blood cultures from patients over 18 years of age with mono-microbial *Enterobacterales* bacteremia. Data on bacterial identification time and AST TAT were extracted from the Laboratory Information System (LIS). In our tertiary care centre, a microbiologist and an infectious diseases practitioner are available from 8:30 a.m. to 6:30 p.m. The microbiologist interprets ASTs between 8:30 a.m. and 6:00 p.m. When a BCB signals positive, the estimated time to MALDI-TOF identification and AST result is 1.5 h, calculated over a 4-month period (1 July 2017 to 31 October 2017) by averaging the time required for bacterial identification directly from a BCB and the time required to perform the AST (there was no change in the routine between 2017 and 2021). The average time required to incubate an MH agar during our study was 16 h. It, therefore, takes 17.5 h to obtain an AST result from a positive BCB. During this four-month period, we estimated the time that could be saved on the AST turnaround time using MHR-SIR compared to the MH medium from the data extracted from our LIS.

### 2.6. Statistical Analysis

The numbers of diameters out of quality control (QC) range for *E. coli* strain ATCC 25922 and AST turnaround times were compared with a paired Student’s *t*-test. The results were significant at a *p*-value < 0.05 (**** *p* < 0.0001). Figures were generated with Rstudio^®^ version 1.3.959 and R^®^ 4.0.1, (RStudio: Integrated Development for R. RStudio, PBC, Boston, MA, USA) and GraphPad Prism version 7. (GraphPad Software, San Diego, CA, USA).

## 3. Results

### 3.1. Comparison of the Performance of AST Directly from BCBs on MHR-SIR Agar Using Two Different Inoculum Protocols Proposed by the BSAC and CASFM-EUCAST, Respectively

Among the 40 ASTs of QC *E. coli* ATCC 25922 read for the repeatability assay, all the means of the diameters were acceptable with the BSAC inoculation protocol, whereas, with the CASFM-EUCAST inoculation protocol, they were below acceptable limits for one antibiotic: ticarcillin (Figure 2). 

Among the 66 ASTs read for the reproducibility assay, all those performed with the BSAC protocol were acceptable. Two antibiotic means were below the acceptable limits using the CASFM-EUCAST protocol (ticarcillin and cefepime) (Figure 3). 

The numbers of disc diameters out of QC range are reported in Table 2. Of the 720 disc diameters analysed for repeatability testing, 19 were outside the QC range stipulated by CASFM-EUCAST 2019 v2 guidelines [14] for *E. coli* strain ATCC 25922 using BSAC inoculum recommendations, while 64 diameters were outside the QC range using the CASFM-EUCAST recommendations. Of the 1188 diameters measured for the reproducibility study, 31 were out of range for AST performed in accordance with BSAC recommendations, and 81 for AST performed according to CASFM-EUCAST recommendations. In total, there were 145 out-of-range diameters (ORD) with the CASFM-EUCAST protocol and 50 with the BSAC protocol (*p* = 0.005). The three antibiotics with the most ORDs were: cefepime (50.9% ORDs using CASFM-EUCAST vs. 24.5% ORDs using BSAC), ticarcillin (39.6 vs. 0%), and ciprofloxacin (34.0 vs. 5.7%). 

In light of these results, we performed AST on MHR-SIR agar following the BSAC inoculum recommendations for the rest of this study. 

Repeatability was achieved with 20 ASTs performed on the same day by the same operator. Reproducibility was performed by different operators and assessed by conducting 33 ASTs

### 3.2. Prospective Comparison of MHR-SIR vs. MH on Enterobacterales Positive Blood Cultures

A total of 110 monomicrobial BCBs from hospitalised patients positive for *Enterobacterales* species were prospectively tested as part of the study. These strains were: *E. coli* (*n* = 51), *Klebsiella pneumoniae* (*n* = 22), *Enterobacter cloacae* (*n* = 13), *Proteus mirabilis* (*n* = 7), *Serratia marcescens* (*n* = 6), *Klebsiella aerogenes* (*n* = 4), *Klebsiella oxytoca* (*n* = 3), *Morganella morganii* (*n* = 2), *Citrobacter koseri* (*n* = 1), and *Proteus hauseri* (*n* = 1). The clinical strains analysed were wild-type for β-lactams (n=72), acquired penicillinase producers (*n* = 18), ESBL producers (*n* = 11), derepressed AmpC mutants (*n* = 5), and inhibitor-resistant TEM producers (*n* = 4). We compared MHR-SIR and MH categorical agreement on 1958 measurements and found only 14 errors (0.7%): 8 me, 5 ME, and 1 VME (Table 3). The MHR-SIR medium, therefore, had 99.3% categorical agreement with the AST reference medium, MH agar. A total of 6 out of 14 errors were for nalidixic acid and ciprofloxacin (2 me and 4 ME), while 1 VME was found for trimethoprim-sulfamethoxazole. ESBL synergy images were also directly detected by the microbiologist on MHR-SIR at day 0 for 9 of the 11 producing strains (82%). For the other 3 remaining strains, a DDST performed on a colony confirmed ESBL production the day after (day 1). 

We then decided to test the ability of the MHR-SIR medium to detect various β-lactam resistance patterns of *Enterobacterales*.

### 3.3. Evaluation of MHR-SIR Ability to Detect MDR Enterobacterales

Of the 39 strains, we were able to detect 8/8 (100%) AmpC hyperproducers, 8/8 (100%) plasmid-mediated cephalosporinase producers, and 10/10 (100%) carbapenemase producers on both MHR-SIR and MH agars. A total of 10 out of 19 (53%) ESBL mechanisms were detected directly by synergy image visualisation on the MHR-SIR medium and 12 out of 19 (63%) on the MH medium. All ESBL producers were identified by DDST on culture isolates on both MHR-SIR and MH media at day 1. Other resistance mechanisms (inhibitor-resistant penicillinase and oxacillinase) were successfully screened (5/5) through interpretation by the microbiologist at day 0. All the resistance mechanisms to β-lactams of the 39 isolates were detected either with or without complementary tests.

Evaluation of the ability of the MHR-SIR medium to detect MDR strains showed there to be 98.2% categorical agreement on 788 measurements with 11 me, 2 ME, and 0 VME. Minor errors concerned the following antibiotics: ticarcillin-clavulanic acid, piperacillin-tazobactam, cefotaxime, ceftazidime, aztreonam, cefepime, and ciprofloxacin. Two ME were observed for SXT. However, MICs measured by E-test on a colony supported the MHR-SIR classification. Interestingly, of the 13 mismatches controlled by MICs, MHR-SIR categorisation was finally considered correct for 77% of them (10/13).

### 3.4. Estimated Reduction in Antibiotic Susceptibility Testing TAT Using Direct MHR-SIR from Positive Blood Cultures

In the next stage, we focused on the reduction in the AST turnaround time, for which we collected data from about 161 BCBs positive for monobacterial *Enterobacterales* during a 4-month period: 78 positive BCBs between 00:00 and 8:30 a.m., 40 between 8:30 a.m. and 3:00 p.m. and 43 between 3:00 p.m. and 00:00. 

The average time required for a reading of the MHR-SIR medium by the automated device was calculated at 7 h. If we include the 1.5 h required for bacterial identification and medium inoculation, 8.5 h were needed to obtain an AST result from a positive BCB.

For a BCB that flags positive between 00:00 and 8:30 a.m. (day 0), the AST on standard MH will be interpreted by a microbiologist the following day (day 1), whereas with the MHR-SIR technology, an AST result will be obtained the same day (day 0). The MHR-SIR protocol, therefore, has a TAT of 8.5 h for *Enterobacterales*-positive BCBs between 00:00 and 8:30 a.m., i.e., 48.5% (78/161) of the blood cultures. When a BCB flags positive between 8:30 a.m. and 3:00 p.m. (day 0), the AST will be read at 8:30 the next morning (day 1) regardless of the type of medium inoculated, meaning no time is gained with the MHR-SIR technology. For BCBs that flag positive between 3:00 p.m. and 0:00 (day 0)*,* i.e., 26.7% (43/161), an AST result will be available the next morning at 8:30 a.m. (day 1) with the MHR-SIR technology, but with the MH protocol, the AST will be read between 9:30 a.m. and 5:30 p.m. (day 1). 

There is, therefore, a time reduction for 75.1% (121/161) of *Enterobacterales* blood cultures with the MHR-SIR method. For those 121 BCBs, the entire AST process with the MH medium takes an average of 27 h, with a minimum of 17.5 h and a maximum of 36.5 h; with the MHR-SIR protocol, it takes an average of 10.8 h, with a minimum of 8.5 h and a maximum of 23.7 h. We calculated a 16-h reduction in the time required to obtain an AST result, which is a significant decrease in AST TAT (**** *p* < 0.0001) (Figure 4).

## 4. Discussion

We have shown here that the performance of MHR-SIR using the BSAC inoculation protocol [12] was comparable to our reference method, MH agar (99.3% categorical agreement), but with the advantage of a reduced AST TAT of 16 h. Moreover, this method can be performed on every *Enterobacterales* for a wide range of antibiotics and also enables all kinds of β-lactam resistance patterns, including AmpC hyperexpression, ESBL, and carbapenemase, to be detected directly from positive BCBs. 

To the best of our knowledge, this is the first study to compare the performance of AST directly from BCBs on MHR-SIR agar using two different inoculation protocols. The analytic performances of the BSAC inoculation protocol were better than the CASFM-EUCAST protocol. Several methods of inoculating agar directly from a positive BCB have already been described, but there is currently no international consensus on any of them [19,20,21,22]. Our results offer evidence in favour of using the MHR-SIR medium in accordance with BSAC recommendations, despite the addition in the EUCAST guidelines of the Area of Technical Uncertainty [9] aiming to minimise systemic and random variations. In our study, two sources of variation can explain the difference in AST results between the CASFM-EUCAST and BSAC inoculum recommendations. Firstly, the disc diameters measured with CASFM-EUCAST standards tended to be smaller for most antibiotics—for ticarcillin and cefepime, they were even out of acceptable limits, which is related to the use of a larger initial inoculum. Secondly, the use of two different media to dilute the blood culture, saline for CASFM-EUCAST and sterile water for BSAC, may have had an impact on bacterial growth, as found in a former study [23]. 

Recently, numerous techniques have been or are being developed to reduce AST TAT [10,24,25,26]. The CLSI recently published recommendations for *Enterobacterales* AST reading in 8 to 10 h [10]. However, there are only 6 antibiotics with breakpoints that can be read (ampicillin, ceftriaxone, ceftazidime, aztreonam, tobramycin, and trimethoprim-sulfamethoxazole). Although 3GC reading is useful for screening ESBL producers and AmpC hyperproducers, the CLSI recommendations do not apply to piperacillin-tazobactam and imipenem, two antibiotics widely used to treat nosocomial sepsis [27,28,29]. EUCAST has published specific diameters depending on incubation time and species for a reading on standard MH after 4, 6, or 8 h of incubation [9,30]. However, this technique is limited to two species of *Enterobacterales*: *E. coli* and *K. pneumoniae*. Moreover, the multiplicity of reading diameters obtained over the incubation time may compromise the implementation of this method in clinical microbiology laboratories. Our results show that integrating the MHR-SIR method into the laboratory workflow would be easier and less constraining. 

AST results obtained directly from BCBs using the MHR-SIR medium at 7 h had remarkable categorical agreement with AST performed with the reference MH medium at 16 h (99.3% in our study). These results are in agreement with previously published work [11]. Perillaud et al. obtained minor and major errors for ofloxacin and ciprofloxacin, confirming our finding that ciprofloxacin is impacted by the 6-h reading on MHR-SIR agar [11,31]. Moreover, ciprofloxacin is one of the antibiotics most affected by the inoculation method: 34.0% of the diameters were out of limits using the CASFM-EUCAST protocol, 5.7% using the BSAC protocol. Our results from the prospective study of MHR-SIR showed two minor errors and four major errors for ciprofloxacin. Nalidixic acid yielded reliable results in our study, and since resistance to this antibiotic predicts resistance development to fluoroquinolones, misclassifications on ciprofloxacin do not have any clinical impact. Isolated resistance to ciprofloxacin (i.e., without nalidixic acid resistance) can be verified by MIC. 

Interestingly, the two previous studies using MHR-SIR technology [11,31] cited amoxicillin-clavulanic acid as displaying VME. However, our results do not confirm this finding as there was categorical agreement between these antibiotics, both prospectively and on collection strains. We found trimethoprim-sulfamethoxazole to be a critical antibiotic with 1 VME and 1 ME in the prospective study and 2 ME in the MDR-strain collection. However, the third method (E-test), calculating MICs, found categorical agreement with the MHR-SIR agar method in three out of four cases, suggesting that MH agar led to the misclassification of trimethoprim-sulfamethoxazole.

Several molecular methods have been developed and used to detect bacterial resistance directly on positive BCBs [32]. Resistance in *Enterobacterales* can be detected by molecular tests targeting multiple resistance genes. The main drawback of these techniques is that they do not detect AmpC hyperproduction and they are very expensive [33]. As of today, phenotypic methods remain the gold standard, allowing testing of the bacterium’s sensitivity to a large panel of antibiotics. In this study, we tested 39 MDR strains on MHR-SIR agar medium in order to assess whether this rapid AST method was able to detect them. To the best of our knowledge, this is the first time that carbapenemase-producing strains were tested on this medium, and we were able to detect 100% of them. 

Directly identifying blood cultures and performing AST 24/7, along with AST interpretation from 8:30 a.m. to 6:00 p.m., are technical and organisational adaptations that can be made to the laboratory workflow to deliver faster AST TAT. In a recent study, the average time saved by MHR-SIR compared with the standard method was estimated at 16 h [34]. At the Nice University Hospital, direct identification of 96% of *Enterobacterales* from BCBs, coupled with reading and interpretation of MHR-SIR AST in an average time of 8.5 h, would deliver a theoretical reduction in TAT of 16 h. This gain has to be re-evaluated in consideration of the human element and workflow organisation. We offer here a precise account of how these factors can be integrated into the workflow of a routine hospital microbiology laboratory. The time saved in the event of an *Enterobacterales* bloodstream infection would shorten the length of hospitalisation [4] and reduce complications and hospital mortality [3]. 

However, our study has several limitations. It is a single-centre study testing only one blood culture system, the BacT/ALERT. It would be of interest to compare these results with other blood culture systems, as another study has shown that categorical agreement can vary according to the system used [35]. Furthermore, a drawback of the MHR-SIR technique is that the AST reading can only be taken by the SIR-scan computer interface (i2a), so it is no longer possible to obtain a precision reading with a calliper. Reading difficulties may therefore arise under poor image acquisition conditions or when the diameters are weakly defined, especially for the detection of ESBL. Based on EUCAST rules, the Combination disk diffusion test (CDT): cefotaxime (30 µg) and ceftazidime (30 µg) ± clavulanic acid (10 µg) and an inhibition zone size difference ≥ 5 mm may be more successful in ESBL detection than DDST at day 0. However, ESBL production was suspected for these two strains on the basis of the reduced disc diameters for 3GC, so double-disc synergy tests were performed, which successfully identified ESBL production at day 1. During testing of the MDR strains, 10 of the 19 (53%) ESBL mechanisms were detected directly by synergy image visualisation on MHR-SIR, and 12 out of 19 (63%) on the MH medium. This small number of synergy visualisations at day 0 may be explained by additional resistance mechanisms carried by the strains, such as AmpC hyperproduction. However, as previously explained, the difficulty in detecting ESBL can be easily overcome with simple complementary tests such as DDST and MH medium supplemented with cloxacillin [15]. Finally, although we obtained a significant reduction in theoretical TAT, this promising result has to be confirmed in practice. 

## 5. Conclusions and Outlook

MHR-SIR agar combined with the SIRscan 2000 Automatic reading system provides a significant reduction in AST incubation time and excellent categorical agreement with disc diffusion AST on MH agar. In collaboration with an infectious diseases department, implementation of this medium will reduce TAT, thus making prompt and efficient antibiotic treatment possible. 

## Figures and Tables

**Figure 1 microorganisms-10-01377-f001:**
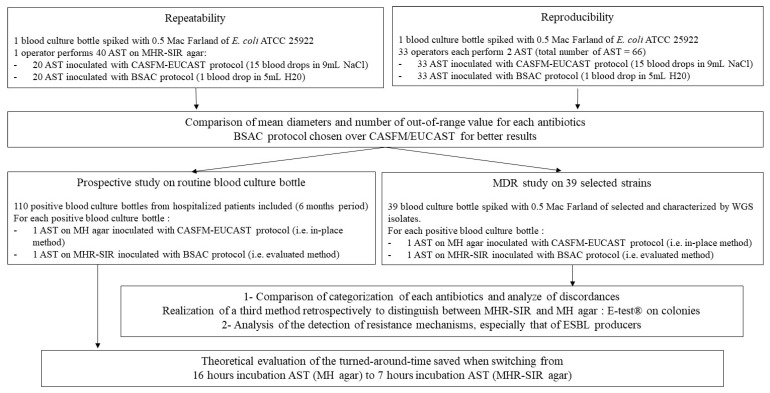
Description of study design.

**Figure 2 microorganisms-10-01377-f002:**
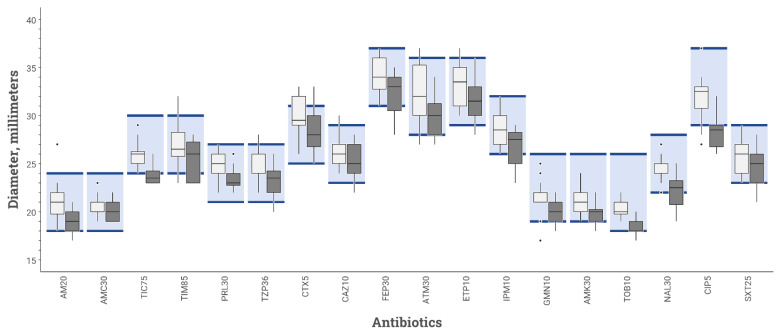
Inhibition diameters for various antibiotics tested on MHR-SIR agar of *E. coli* strain ATCC 25922 obtained during repeatability study comparing 2 inoculation protocols: BSAC (white box-plot) vs. CASFM-EUCAST (black box-plot) recommendations. The boxplot shows the 25th and 75th percentile with a horizontal bar indicating the median. Whiskers represent the 5th and 95th percentile. Extreme values are shown with dots. The colored rectangle indicates the acceptable limit values for each antibiotic according to the CASFM 2019 v2 recommendations. AM20 = Amoxicillin 20 µg, AMC30 = Amoxicillin-clavulanic acid 30 µg, TIC75 = Ticarcillin 75 µg, TIM85 = Ticarcillin-clavulanic acid 85 µg, PRL30 = Piperacillin 30 µg, TPZ36 = Piperacillin-tazobactam 36 µg, CTX5 = Cefotaxime 5 µg, CAZ10 = Ceftazidime 10 µg, FEP30 = Cefepime 30 µg, ATM30 = Aztreonam 30 µg, ETP10 = Ertapenem 10 µg, IPM10 = Imipenem 10 µg, GN10 = Gentamicin 10 µg, AK30 = Amikacin 30 µg, TOB10 = Tobramycin 10 µg, NA30 = Nalidixic acid 30 µg, CIP5 = Ciprofloxacin 5 µg, SXT25 = Trimethoprim + sulfamethoxazole 25 µg.

**Figure 3 microorganisms-10-01377-f003:**
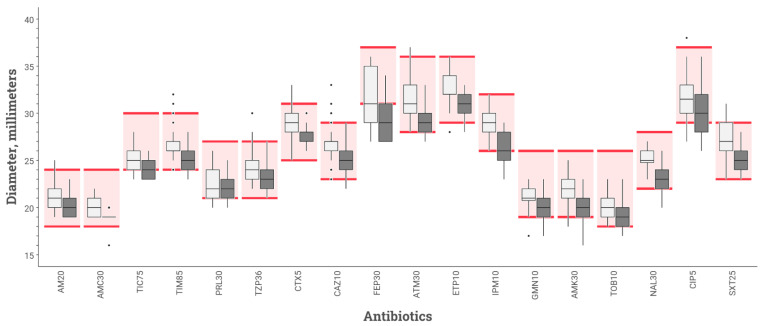
Inhibition diameters for various antibiotics tested on MHR-SIR agar of *E. coli* strain ATCC 25922 obtained during reproducibility study comparing 2 inoculation protocols: BSAC (white box-plot) vs. CASFM-EUCAST (black box-plot) recommendations. The boxplot shows the 25th and 75th percentile with a horizontal bar indicating the median. Whiskers represent the 5th and 95th percentile. Extreme values are shown with dots. The colored rectangle indicates the acceptable limit values for each antibiotic according to the CASFM 2019 v2 recommendations. AM20 = Amoxicillin 20 µg, AMC30 = Amoxicillin-clavulanic acid 30 µg, TIC75 = Ticarcillin 75 µg, TIM85 = Ticarcillin-clavulanic acid 85 µg, PRL30 = Piperacillin 30 µg, TPZ36 = Piperacillin-tazobactam 36 µg, CTX5 = Cefotaxime 5 µg, CAZ10 = Ceftazidime 10 µg, FEP30 = Cefepime 30 µg, ATM30 = Aztreonam 30 µg, ETP10 = Ertapenem 10 µg, IPM10 = Imipenem 10 µg, GN10 = Gentamicin 10 µg, AK30 = Amikacin 30 µg, TOB10 = Tobramycin 10 µg, NA30 = Nalidixic acid 30 µg, CIP5 = Ciprofloxacin 5 µg, SXT25 = Trimethoprim + sulfamethoxazole 25 µg.

**Figure 4 microorganisms-10-01377-f004:**
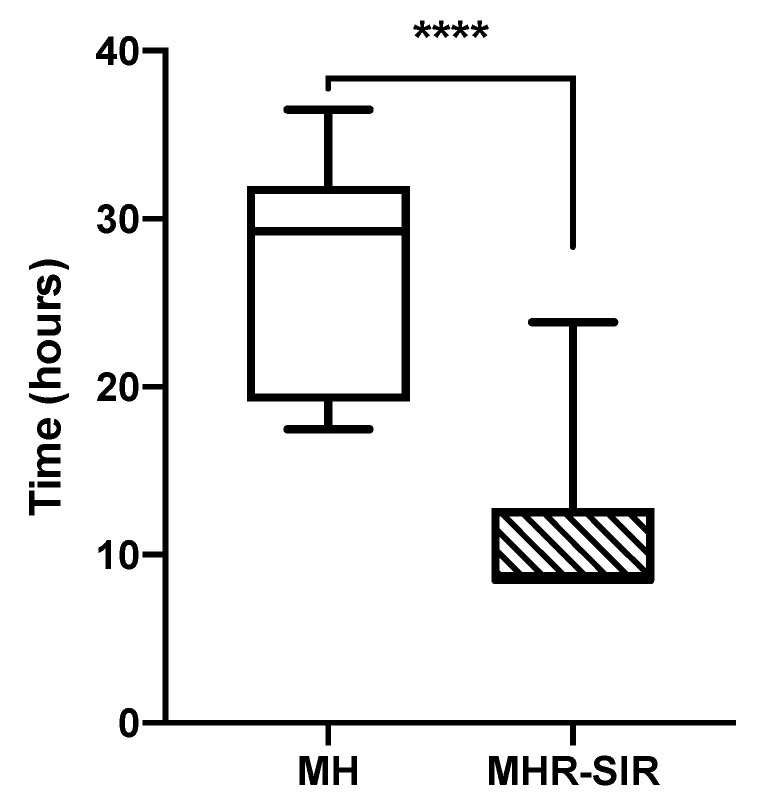
Comparison of time required to obtain the direct AST from a blood culture bottle for MH vs. MHR-SIR (**** *p* < 0.0001). Boxplots show the 25th and 75th percentile, with the horizontal bar showing the median. Whiskers show the 5th and 95th percentile. The outliers are represented by dots.

**Table 1 microorganisms-10-01377-t001:** β-lactam resistance mechanisms of 39 selected *Enterobacterales* isolates evaluated with MHR-SIR and MH agar.

Bacteria	Genetic β-Lactam Resistance *	Phenotypic Mechanism
*E. coli*	*NDM-5*, *OXA-181*, *CMY-42*	carbapenemase
*E. coli*	*TEM-1*, *OXA-48*	carbapenemase
*E. coli*	*OXA-48*	carbapenemase
*K. pneumoniae*	*OXA-1*, *TEM-1*, *NDM-5*	carbapenemase
*K. pneumoniae*	*CTX-M-15*, *NDM-1*, *CMY-2*, *OXA-1*	carbapenemase, ESBL
*K. pneumoniae*	*CTX-M-15*, *NDM-1*, *TEM-1*, *OXA-1*, *OXA-9*	carbapenemase, ESBL
*E.cloacae*	*CTX-M-15*, *KPC-3*	carbapenemase, ESBL
*C. freundii*	*CTX-M-15*, *OXA-48*, *OXA-1*, *OXA-9*, *AmpC hyperproducter*	carbapenemase, ESBL, AmpC hyperproduction
*C. freundii*	*CTX-M-15*, *OXA-48*, *AmpC hyperproducter*	carbapenemase, ESBL, AmpC hyperproduction
*E.cloacae*	*SHV-12*, *AmpC hyperproducter*, *OXA-48*	carbapenemase, ESBL, AmpC hyperproduction
*K. oxytoca*	*SHV-12*, *TEM-1*	ESBL
*E. coli*	*CTX-M-27*	ESBL
*K. pneumoniae*	*CTX-M-15*, *TEM-1*, *OXA-1*, *OXA-9*	ESBL
*E. coli*	*CTX-M-27*, *TEM-1*	ESBL
*E. coli*	*CTX-M-15*, *OXA-1*	ESBL
*E. coli*	*CTX-M-27*	ESBL
*K. pneumoniae*	*CTX-M-15*	ESBL
*E. coli*	*CTX-M-14*	ESBL
*E. coli*	*CTX-M-14*	ESBL
*E. coli*	*CTX-M-27*	ESBL
*E. coli*	*CTX-M-15*, *TEM-1*	ESBL
*E. coli*	*CTX-M-15*, *TEM-30*, *OXA-1*	ESBL
*E. cloacae*	*SHV-12*, *TEM-1*, *AmpC hyperproducter*	ESBL, AmpC hyperproducer
*E. cloacae*	*AmpC*	AmpC hyperproduction
*E. cloacae*	*AmpC*	AmpC hyperproduction
*E. coli*	*AmpC*	AmpC hyperproduction
*E. coli*	*AmpC*	AmpC hyperproduction
*S. marcescens*	*AmpC*	AmpC hyperproduction
*K. oxytoca*	*DHA-1 CMY-2*	plasmidic cephalosporinase
*P. mirabilis*	*CMY-2*	plasmidic cephalosporinase
*E. coli*	*CMY-2*	plasmidic cephalosporinase
*E. coli*	*DHA-1 TEM-1*	plasmidic cephalosporinase
*K. pneumoniae*	*DHA-1 TEM-1*	plasmidic cephalosporinase
*K. pneumoniae*	*DHA-1*	plasmidic cephalosporinase
*K. pneumoniae*	*SHV-27 + TEM-1*	Penicillinase hyperproduction
*K. pneumoniae*	*SHV-1*	Penicillinase hyperproduction
*P. mirabilis*	*HyperTEM-1 (Pa*/*Pb)*	oxacillinase
*E. coli*	*TEM-33*	Inhibitor-resistant TEM
*E. coli*	*OXA-1*	oxacillinase

* obtained by whole-genome sequencing analysis.

**Table 2 microorganisms-10-01377-t002:** Numbers of out-of-range diameters for quality control *E. coli* strain ATCC 25922 with two MHR-SIR inoculation protocols: CASFM-EUCAST (15 drops of blood in 9 mL NaCl) vs. BSAC (1 drop of blood in 5 mL H_2_O).

Antibiotics	Numbers of Out-of-Range Diameters			
	CASFM-EUCAST		BSAC	
	Repeatability	Reproducibility	Repeatability	Reproducibility
Amoxicillin 20 µg	3	0	1	0
Amoxicillin-clavulanic acid 30 µg	0	1	0	0
Ticarcillin 75 µg	10	11	0	0
Ticarcillin-clavulanic acid 85 µg	6	2	1	2
Piperacillin 30 µg	0	8	0	4
Piperacillin-tazobactam 36 µg	1	0	1	3
Cefotaxime 5 µg	1	0	6	1
Ceftazidime 10 µg	2	1	1	3
Cefepime 30 µg	5	22	0	13
Aztreonam 30 µg	3	4	4	1
Ertapenem 10 µg	2	1	2	0
Imipenem 10 µg	6	8	0	0
Gentamicin 10 µg	0	4	1	1
Amikacin 30 µg	3	3	0	1
Tobramycin 10 µg	3	3	0	0
Nalidixic acid 30 µg	6	4	0	0
Ciprofloxacin 5 µg	9	9	2	1
Trimethoprim-sulfamethoxazole 25 µg	4	0	0	1
Total	64	81	19	31

**Table 3 microorganisms-10-01377-t003:** Antibiotic categorisation discordances between ASTs performed on MHR-SIR and on MH agar from 110 *Enterobacterales*-positive blood culture bottles.

Bacterium	β-Lactams Resistance	Type of Discordance	Molecule	MIC (mg/L) E-Test Method	Correct Method
*K. pneumoniae*	ESBL producer	VME	trimethoprim-sulfamethoxazole	32 (R)	MH
*E. coli*	WT	ME	ciprofloxacin	0.19 (S)	MH
*E. coli*	WT	ME	ciprofloxacin	0.006 (S)	MH
*E. coli*	Penicillinase	ME	ciprofloxacin	0.5 (I)	None
*K. pneumoniae*	WT	ME	trimethoprim-Sulfamethoxazole	0.032 (S)	MHR-SIR
*S. marcescens*	WT	ME	ciprofloxacin	0.064 (S)	MH
*E. coli*	Penicillinase	me	trimethoprim-sulfamethoxazole	32 (R)	MH
*E. coli*	ESBL producer	me	ciprofloxacin	0.19 (S)	None
*K. pneumoniae*	WT	me	ticarcillin-clavulanic acid	16 (I)	MHR-SIR
*K. pneumoniae*	ESBL producer	me	piperacillin-tazobactam	48 (R)	MHR-SIR
*K. pneumoniae*	ESBL producer	me	aztreonam	6 (R)	MH
*P. mirabilis*	WT	me	ciprofloxacin	0.5 (I)	MH
*E. cloacae*	AmpC hyperproducer	me	piperacillin-tazobactam	64 (R)	MH
*E. cloacae*	AmpC hyperproducer	me	aztreonam	12 (R)	MHR-SIR

WT = Wild type, VME = very major error, ME = major error, me = minor error, MH = Muller-Hinton, MHR = Mueller-Hinton Rapid-SIR, MIC = minimum inhibitory concentration.

## Data Availability

Data are available upon reasonable request.

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
