# Peer review of "Evaluation of an Antibiotic Susceptibility Testing Method on Enterobacterales-Positive Blood Cultures in Less Than 8 h Using the Rapid Mueller-Hinton Diffusion Method in Conjunction with the SIRscan 2000 Automatic Reading Device"

_microorganisms, 2022, doi:10.3390/microorganisms10071377_

Round 1

Reviewer 1 Report

The manuscript is interesting and certainly focuses on an important issue which is to accelerate the detection of antibiotic resistance to improve the treatment of patients. From my point of view the material and methods section is a bit complex as it is described and does not make it easy for the reader to understand. So this is a point that should be reviewed by the authors.

Other comments:

In material and methods sections please revise that all the commercial brands of all the reagents used are indicated. For example

Page 6 section 2.4: The whole genome sequencing method is not properly described. How the DNA was isolated? What kit was used for library preparation? Equipment used for sequencing?

Author Response

The manuscript is interesting and certainly focuses on an important issue which is to accelerate the detection of antibiotic resistance to improve the treatment of patients. From my point of view the material and methods section is a bit complex as it is described and does not make it easy for the reader to understand. So this is a point that should be reviewed by the authors.

We thank the reviewer 1 for this analysis of our study. We agree that the material and methods section is a bit complex but since that the SIRscan 2000 automatic incubator-reader is a device which is not extensively used in all laboratories, it seems important to give the main details of this system. Moreover, the readers of this manuscript must be able to reproduce the methods performed. However, we reduced some sentences in the first paragraph of material methods section.  

Other comments:

In material and methods sections please revise that all the commercial brands of all the reagents used are indicated. For example

Page 6 section 2.4: The whole genome sequencing method is not properly described. How the DNA was isolated? What kit was used for library preparation? Equipment used for sequencing?

We agree with these two comments made by the reviewer 1. We revised that all the commercial brands of all reagents used are indicated and have properly described the whole genome sequencing method as suggested by the reviewer

Reviewer 2 Report

Overall this is a well-conducted study, highlighting advantages and disadvantages of the recently-developed MHR-SIR medium for performing AST by using the SIRscan 200 Automatic system. Data are clearly presented, even if efforts should be done to condense the size of the manuscript and particularly shorten the Discussion section.

One major issue is the lack of MIC data that unable the reader to figure out what is the "real" phenotype of some resistant isolates. For instance, are those OXA-48 producers included in the study showing high-level resistance to carbapenems, or are they (as it is often the case) showing a borderline resistance to them ? Those data must be included for sake of transparency. Same applies with cefotaxime or ceftazidime for the ESBL producers.

Detection of carbapenemases; was that performed with the Carba NP test or the NitroSpeed Carba NP test as suggested by the quoted reference (ref. 18) ? 

Please clarify in Table 2 the what is exactly meant by "repeatability" and by "reproducibility".

Author Response

Overall this is a well-conducted study, highlighting advantages and disadvantages of the recently-developed MHR-SIR medium for performing AST by using the SIRscan 200 Automatic system. Data are clearly presented, even if efforts should be done to condense the size of the manuscript and particularly shorten the Discussion section.

We shortened the discussion section and to a lesser degree the material and methods section

One major issue is the lack of MIC data that unable the reader to figure out what is the "real" phenotype of some resistant isolates. For instance, are those OXA-48 producers included in the study showing high-level resistance to carbapenems, or are they (as it is often the case) showing a borderline resistance to them ? Those data must be included for sake of transparency. Same applies with cefotaxime or ceftazidime for the ESBL producers.

As suggested by the reviewer, we determined the MIC of carbapenems (Imipenem and Meropenem), cefotaxime and ceftazidime for all 39 strains of our collection including carbapenemases producers and ESBL producers. (see results in supplementary Table)

Detection of carbapenemases; was that performed with the Carba NP test or the NitroSpeed Carba NP test as suggested by the quoted reference (ref. 18) ? 

We corrected the reference about the Carba NP test. Indeed, the correct reference is : Nordmann P, Poirel L, Dortet L. 2012. Rapid detection of carbapenemase-producing Enterobacteriaceae. Emerg Infect Dis 18:1503–1507.

Please clarify in Table 2 the what is exactly meant by "repeatability" and by "reproducibility".

We have specify the term of «repeatability" and by "reproducibility " in Table 2 footnote

Repeatability was achieved with 20 ASTs performed on the same day by the same operator

Reproducibility was performed by different operators, was assessed by conducting 33 ASTs

Reviewer 3 Report

Corrections needed:

1.     Page 3, trimethoprim-Sulfamethoxazole 25 μg: small letter

2.     Page 4, E. coli: italics

3.     Page 5, BLSE: ESBL? or ß-lactamase? unclear and/or unexplained abbreviation

4.     5.     Page 7, SHV-27: is ESBL and this is greater resistance than penicillinase hyperproduction

6.     Page 11, Enterobacter aerogenes: newly named Klebsiella aerogenes

7.     Page 11, (: unnecessary bracket

8.     Page 12, WT: unexplained abbreviation; Without Testing?

9.     Page 14, µl: µL

10.    Page 15, ...synergy images were not visualised directly on the MHR-SIR medium for 2 out of 11 ESBL producers (18%).: Based on EUCAST rules, the Combination disk diffusion test (CDT): cefotaxime (30 µg) and ceftazidime (30 µg) ± clavulanic acid (10 µg) and an inhibition zone size difference ≥ 5 mm may be more successful in ESBL detection than DDST at day 0. As you say later, ESBL identification may be influenced by additional resistance mechanisms such as AmpC  hyperproduction/or plasmid mediated ß-lactamase and carbapenemase.

Author Response

Reviewer 3

Comments and Suggestions for Authors

Corrections needed:

  1. Page 3, trimethoprim-Sulfamethoxazole 25 μg: small letter

We corrected the typography of sulfamethoxazole

  1. Page 4,  coli: italics

We wrote E. coli in italics

  1. Page 5, BLSE: ESBL? or ß-lactamase? unclear and/or unexplained abbreviation We corrected BLSE as ESBL and specified abbreviation ESBL : extended spectrum ß-lactamase
  2.    5.     Page 7, SHV-27: is ESBL and this is greater resistance than penicillinase hyperproduction

Corkill et al. (PMID: 11266422) have originally published that SHV-27 as a novel cefotaxime-hydrolysing beta-lactamase in Klebsiella pneumoniae. However, they did not confirm this result by cloning this gene or by enzymatic characerization. Moreover, they have not done CTX-M by PCR.

Actually, a more recent paper has confirmed that SHV-27 was not a ESBL but only a penicillinase. This error remained in the databases since that the article was not retracted or corrected

Extended-spectrum beta-lactamase genes of Klebsiella pneumoniae strains in Taiwan: recharacterization of shv-27, shv-41, and tem-116.

Lin TL, Tang SI, Fang CT, Hsueh PR, Chang SC, Wang JT.Microb Drug Resist. 2006 Spring;12(1):12-5. doi: 10.1089/mdr.2006.12.12.PMID: 16584302

  1. Page 11, Enterobacter aerogenes: newly named Klebsiella aerogenes

We corrected the new name of Enterobacter aerogenes 

  1. Page 11, (: unnecessary bracket

We deleted the bracket

  1. Page 12, WT: unexplained abbreviation; Without Testing?

We give explanation of the abbreviation WT is Wild Type

  1. Page 14, µl: µL

We found and corrected µl in Page 15

  1. Page 15, ...synergy images were not visualised directly on the MHR-SIR medium for 2 out of 11 ESBL producers (18%).: Based on EUCAST rules, the Combination disk diffusion test (CDT): cefotaxime (30 µg) and ceftazidime (30 µg) ± clavulanic acid (10 µg) and an inhibition zone size difference ≥ 5 mm may be more successful in ESBL detection than DDST at day 0. As you say later, ESBL identification may be influenced by additional resistance mechanisms such as AmpC  hyperproduction/or plasmid mediated ß-lactamase and carbapenemase.

We agree with this comment made by the reviewer 3 and insert this remark in the discussion section